# Spin-polarized surface resonances accompanying topological surface state formation

Chris Jozwiak[1,*], Jonathan A. Sobota[1,2,3,*], Kenneth Gotlieb[4], Alexander F. Kemper[5,6], Costel R. Rotundu[2], Robert J. Birgeneau[7,8,9], Zahid Hussain[1], Dung-Hai Lee[7,8], Zhi-Xun Shen[2,3] & Alessandra Lanzara[7,8]

Topological insulators host spin-polarized surface states born out of the energetic inversion of bulk bands driven by the spin-orbit interaction. Here we discover previously unidentified consequences of band-inversion on the surface electronic structure of the topological insulator $Bi_2Se_3$. By performing simultaneous spin, time, and angle-resolved photoemission spectroscopy, we map the spin-polarized unoccupied electronic structure and identify a surface resonance which is distinct from the topological surface state, yet shares a similar spin-orbital texture with opposite orientation. Its momentum dependence and spin texture imply an intimate connection with the topological surface state. Calculations show these two distinct states can emerge from trivial Rashba-like states that change topology through the spin-orbit-induced band inversion. This work thus provides a compelling view of the coevolution of surface states through a topological phase transition, enabled by the unique capability of directly measuring the spin-polarized unoccupied band structure.

[1] Advanced Light Source, Lawrence Berkeley National Laboratory, Berkeley, California 94720, USA. [2] Stanford Institute for Materials and Energy Sciences, SLAC National Accelerator Laboratory, 2575 Sand Hill Road, Menlo Park, CA 94025, USA. [3] Geballe Laboratory for Advanced Materials, Departments of Physics and Applied Physics, Stanford University, Stanford, CA 94305, USA. [4] Graduate Group in Applied Science and Technology, University of California, Berkeley, California 94720, USA. [5] Department of Physics, North Carolina State University, Raleigh, North Carolina 27695, USA. [6] Computational Research Division, Lawrence Berkeley National Laboratory, Berkeley, California 94720, USA. [7] Materials Sciences Division, Lawrence Berkeley National Laboratory, Berkeley, California 94720, USA. [8] Department of Physics, University of California, Berkeley, California 94720, USA. [9] Department of Materials Science and Engineering, University of California, Berkeley, California 94720, USA. * These authors contributed equally to this work. Correspondence and requests for materials should be addressed to C.J. (email: CMJozwiak@lbl.gov) or to A.L. (email: ALanzara@lbl.gov).

The remarkable progress of theoretical and experimental studies on topological insulators has dramatically deepened our understanding of how the spin-orbit interaction (SOI) can shape electronic bandstructure in solids. It is now well-established that band inversion driven by SOI is responsible for the formation of a topological surface state (TSS) in a range of materials[1–4]. The underlying influence of SOI in these states can be directly studied by spin- and angle-resolved photoemission spectroscopy (ARPES), which unveils not only the characteristic helical spin structure[5–11], but also the more intricate coupling of orbital and spin textures[12–14]. The latter can be probed through the direct correlation between the spin orientation of the outgoing photoelectron and the polarization vector of the incident photon[15–17]. The extent of the impact of SOI in these materials has been further evidenced by the observation of a second TSS existing well above the Fermi level[18,19].

Here we provide new insights on SOI-driven band inversion by employing spin, time, and angle-resolved photoemission spectroscopy (STARPES) to study the spin polarization of transiently occupied states above the Fermi energy, $E_F$ in the prototypical topological insulator $Bi_2Se_3$ through optical pump excitation. These experiments were performed using a high-efficiency and high-resolution spin-resolved photoelectron spectrometer[20,21], which is critical for enabling polarization analysis on states far above the Fermi energy, $E_F$, that have inherently low signal strength. We observe a spin-polarized unoccupied surface resonance (USR) within the bulk conduction band (BCB), which is distinct from the TSS[22,23] and has a helical spin texture opposite to that of the TSS. Moreover, it is characterized by entangled spin and orbital textures similar to that of the TSS[12–14], manifested by the measured dependence of the orientation of the photoelectron spin polarization on the photon polarization[15–17]. Our observation of the USR is corroborated by density functional theory (DFT) calculations, and tight-binding calculations provide plausible scenarios in which the spin textures of the USR and TSS are intimately related as the two coevolve from a pair of Rashba-like states through the SOI band inversion.

## Results

**Spin resolved electronic structure above $E_F$.** The purpose of this paper is to probe the spin texture of the USR in detail and to elucidate its relationship with the TSS. A schematic of the experimental geometry is shown in Fig. 1a, with the expected band structure sketched in Fig. 1b. Figure 1c,d presents spin-integrated ARPES data on $Bi_2Se_3$ taken along the Γ-K direction. The equilibrium (unpumped) spectrum (panel (c)) shows the BCB and TSS, both terminated at $E_F$[24,25]. At a delay $\Delta t = 0.7$ ps after optical excitation with 1.5 eV photons (panel (d)), there is a depletion of population below $E_F$ with a concomitant increase above, consistent with previous investigations[26–31]. To increase the visibility of higher energy states, the spectrum in (d) has been normalized by a Fermi-Dirac distribution with elevated electronic temperature ($k_B T = 85$ meV).

The main observation is a continuation of the TSS, linearly dispersing far above the band gap and $E_F$, remaining distinct from the BCB. Interestingly, we observe an increase in spectral weight toward the top of the BCB (beyond $k_x \approx 0.1 Å^{-1}$) which we identify as a USR in accordance with previous works[22,23]. We show in Supplementary Fig. 1 that this observation is not dependent on the Fermi-Dirac normalization.

The unique capabilities of STARPES allow us to measure not only the spectral weight, but also the spin-polarization of the bands above $E_F$. The STARPES spectrum is shown in Fig. 1e with a method of visualizing extensive spin-resolved data, where a two-dimensional colorscale is used to represent both spin

polarization and spectral intensity (see Supplementary Fig. 2 for more details of this plotting method). The dominant feature is the spin-polarization of the TSS, as has been observed previously in equilibrium below $E_F$[5–11]. Here we see that the TSS maintains its helical spin texture well above $E_F$. We further find that the USR is characterized by a spin-polarization in a direction opposite to that of the TSS. Because bulk states are necessarily spin-degenerate due to the inversion symmetry of the crystal, the measured spin polarization of the USR is consistent with it having a strong surface-localized component. Analysis of the data in Supplementary Fig. 3 suggests that the polarization of the USR vanishes gradually toward Γ, as opposed to the TSS which maintains its polarization all the way to the immediate vicinity of $k = 0$. These measurements present a striking contrast from the typical schematic of the topological insulator electronic structure sketched in Fig. 1b; not only because of the existence of the USR, but also because the upper branches of the TSS extend far beyond the bulk gap, continuing well above the bottom of the BCB and well outside the region of band inversion. These measurements suggest an interesting relationship between the TSS and USR.

Motivated by these observations, we now investigate the properties of the USR more systematically. Figure 2a presents both equilibrium and pumped spin resolved spectra, collected at a fixed emission angle slightly beyond $k_F$, corresponding to the vertical dashed line in Fig. 1c–e. The equilibrium spectrum is strongly spin-polarized due to its proximity to the TSS, with its intensity strongly diminished beyond $E_F$. The spin-polarization is observed up to $E - E_F \approx 0.2$ eV due to the thermal population above $E_F$ at the sample temperature of 80 K; beyond this energy, the population is too low to obtain a statistically significant polarization value. This limitation is alleviated by the transient occupation of states above $E_F$ created by the pump excitation. The pumped spectrum shows the spectral peak of the TSS now well above $E_F$ at $\approx 0.1$ eV. With the increased measurement range we also see a reversal of the spin-polarization beyond $E - E_F \approx 0.3$ eV, and a corresponding switch to the intensity spectrum being dominated by spin-up photoelectrons (see Fig. 2b). These features, marked by arrows, are associated with the USR as identified in Fig. 1.

We also measured the spectrum with a $p$-polarized probe beam, shown in Fig. 2c, which causes the photoelectron spin-polarization to systematically reverse throughout the measured energy window. This shows that the probe-polarization-dependence, previously observed for the TSS and interpreted in the context of entangled spin-orbit textures[15–17], applies also to the USR. In contrast, the polarization of the pump has no effect on the measured spin-polarization, as shown in Supplementary Fig. 4.

**Time independence of spin polarizations.** Before interpreting the measured photoelectron spin polarizations above $E_F$ in terms of the spin texture of the USR equilibrium eigenstates, one must first rule out the possibility that the measured polarization is due to the pump excitation selectively occupying these states with electrons of a particular spin orientation, for instance due to spin-dependent matrix elements[32]. If the transiently occupied eigenstates were in fact spin unpolarized, such a population would tend to depolarize on a sub-picosecond timescale[33]. Therefore, one method to exclude this scenario is to study the dynamics of the polarization.

In Fig. 3a,b, we present the time dependence of the total (spin-integrated) intensity and corresponding spin polarization, respectively, of the same spectrum as in Fig. 2 and corresponding to the vertical dashed lines in Fig. 1c–e. The polarization plot has three predominant energy regions corresponding to the negatively polarized TSS (red), the unpolarized BCB (white), and

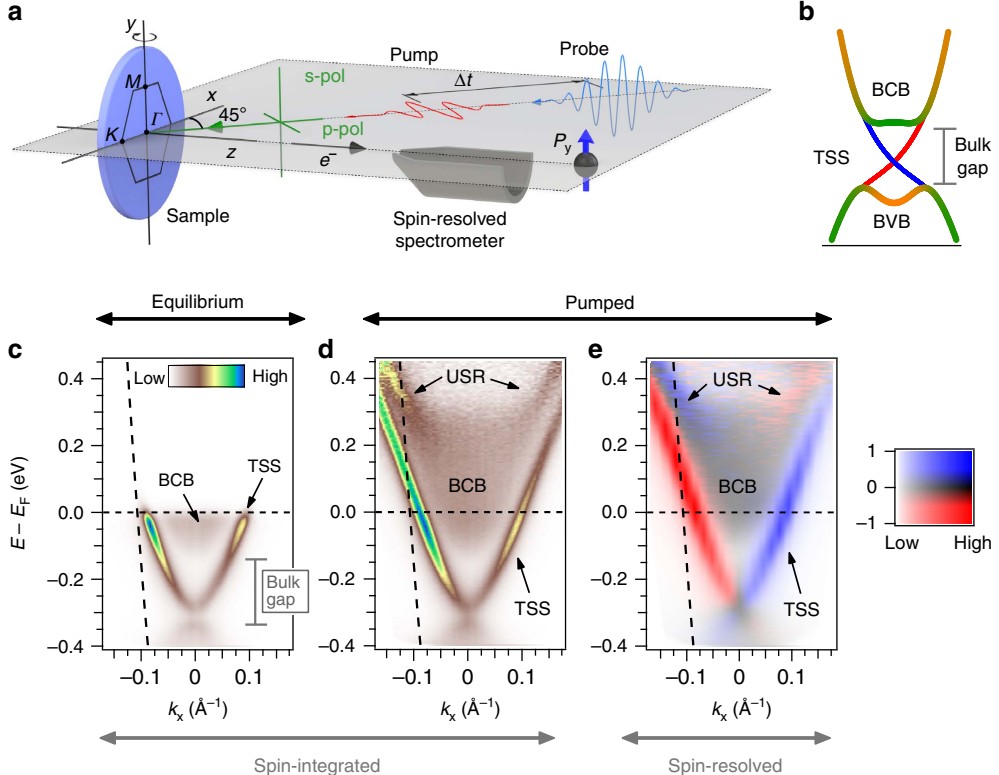

**Figure 1 | Spin resolved electronic structure of Bi₂Se₃ below and above the Fermi level.** (**a**) Experimental geometry of the spin, time, and angle-resolved photoemission measurement. 1.5 eV pump and 6 eV probe pulses are incident on the sample with relative time delay $\Delta t$. Unless otherwise specified, the pump is $p$-polarized and the probe is $s$-polarized. The orientation of the crystal axes with respect to the pulses and spin-resolved spectrometer is as depicted for data taken along Γ-K. Angle-resolved spectra are obtained by rotating the sample with respect to the $y$-axis. (**b**) A naive expectation for the electronic structure of the topological surface state (TSS) formed by spin-orbit-induced band inversion. The characteristic M-shape of the bulk valence band (BVB) is a signature of its hybridization with the bulk conduction band (BCB) driven by band inversion. The TSS exists within the bulk gap, within the band inversion region. (**c**) Equilibrium (no pump) spectrum taken along Γ-K in spin-integrated mode. The colorscale represents photoemission intensity. (**d**) Same spectrum, but with pump, at a time delay $\Delta t = 0.7$ ps. Here, the spectrum is also divided by the effective Fermi-Dirac distribution ($k_B T = 85$ meV) of the pumped spectrum to allow the resolved features through the wide intensity dynamic range of the spectrum to be visible in a single color scale. The continuation of the TSS above the bulk band gap is well resolved. An increase in spectral weight toward the top of the BCB coincides with the unoccupied surface resonance (USR). (**e**) Spin-resolved map of the pumped spectrum, showing that the USR is spin-polarized in a direction opposite to that of the TSS. The spectrum is plotted with the two-dimensional colorscale shown, with the horizontal axis corresponding to intensity and the vertical axis corresponding to spin polarization.

the positively polarized USR (blue). In Fig. 3c,d we extract the delay-dependent intensity and polarization of the TSS (above $E_F$) and the USR, integrating through the purple and green boxes, respectively, in panels (a) and (b). Despite the substantial increase of population due to pumping, the polarization within the purple box is constant for all delays, evidencing the robustness of the spin texture of the TSS. The behavior within the green box is precisely the same: despite substantial population changes, the polarization is constant within experimental error bars. This time-independence suggests that the measured polarizations of both the USR and TSS are attributable to spin polarization of the eigenstates rather than transient polarizations induced by the pump excitation.

**DFT calculation of the electronic structure.** Our characterization of the USR can be further justified by theoretical considerations. We performed DFT calculations of the electronic structure of a 7-quintuple layer Bi₂Se₃ slab, shown in Fig. 4a. The usual TSS is clearly observed as a linearly-dispersive, spin-polarized band which traverses the gap between the bulk valence band (BVB) and BCB. We find states within a certain interval of $k$-space away from Γ that are derived from the BCB yet

exhibit a strong spin-polarization. The corresponding wavefunctions exhibit a strong component localized to the surface, and thus are justified to be called a surface resonance. Consistent with our measurements, the DFT calculated spin-polarization of the USR is opposite to that of the TSS. Moreover, the DFT results show that the USR has a coupled spin-orbital texture like that of the TSS[13,14,16], but with reversed spin textures for each orbital projection, as shown in Supplementary Fig. 5. This is demonstrated by Fig. 4b–d, in which the electronic structure is projected onto the $p_z$, $p_x$, and $p_y$ orbital contributions, respectively. Photoemission in the present geometry (Fig. 1a) preferentially photoemits from states of $p_y$ orbital symmetry with $s$-polarized light, and from states of $p_x$ and $p_z$ orbital symmetries with $p$-polarized light. Therefore, the spin-polarized map reproduced in Fig. 4e, measured with $s$-polarized light, can be compared with the $p_y$ projected electronic structure (panel (d)), with remarkable agreement. This is also consistent with the measured photon polarization-dependent spin-reversal (Fig. 2c), as the spin orientations calculated in Fig. 4d are opposite from that of panels (b,c) for both TSS and USR.

We note that the USR has had experimental signatures in other probes of topological insulators. It was first identified in calculations of Bi₂Te₂Se, where it was invoked to explain

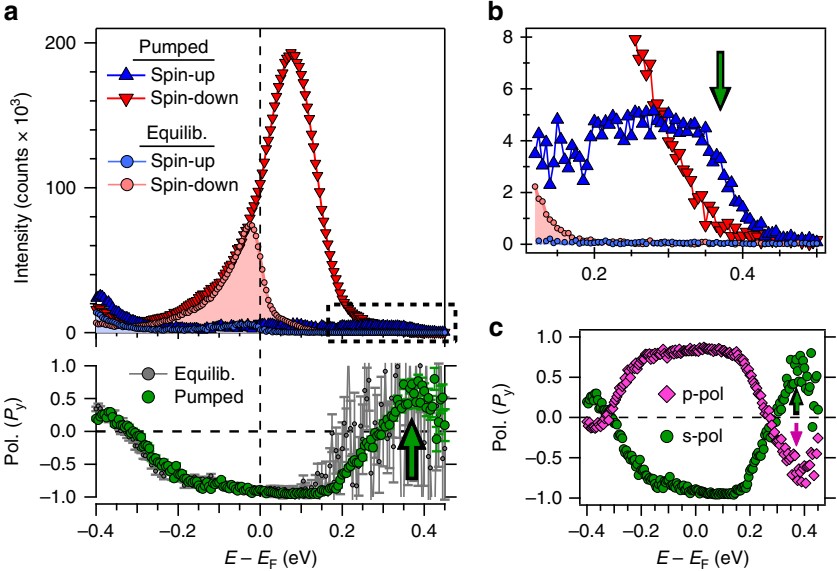

**Figure 2 | Spin resolved spectra at a fixed emission angle beyond $k_F$.** (**a**) The pump excitation allows spin analysis to be performed well above the Fermi energy $E_F$. The error bars shown in the spin polarization $P_y$ are statistical error bars, calculated as $\Delta P_y = \left(S_{eff}\sqrt{N}\right)^{-1}$, where $S_{eff} = 0.2$ is the effective Sherman function of the spectrometer and $N$ is the total number of recorded counts. The error bars for the equilibrium measurement (gray) increase rapidly above $E_F$ due to the low number of counts from the low occupation in equilibrium. Only once states above $E_F$ are populated through pumping can a reasonable measurement be made. (**b**) Upon zooming in, it is apparent that polarization ($P_y$) reverses at $E - E_F \approx 0.3$ eV within the pumped ($\Delta t = 0.7$ ps) spectrum. (**c**) The measured spin-polarization reverses sign when measured with $s$- and $p$- polarized probe photons. In both cases the pump is $p$-polarized.

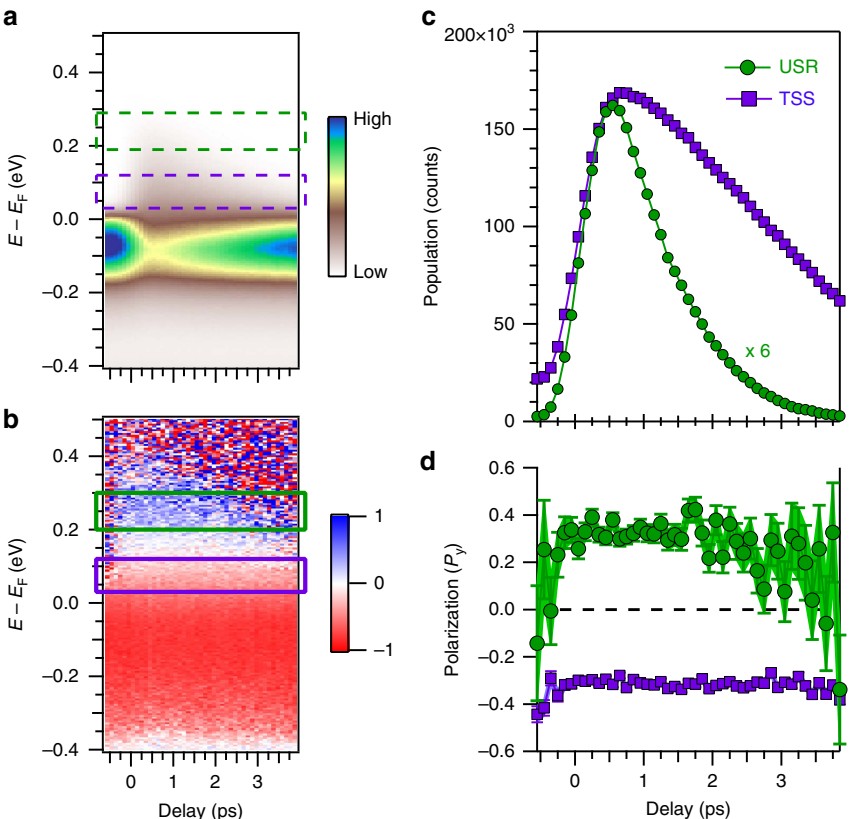

**Figure 3 | Time independence of the spin polarization.** (**a**) The spin-integrated spectrum plotted versus pump-probe delay and (**b**) its corresponding spin-polarization. The time dependence is obtained by changing the delay between the arrival of the pump and probe pulses on the sample surface. Two energy regions corresponding to the unoccupied surface resonance (USR) and topological surface state (TSS) are demarcated with boxes. The colorscales represent photoemission intensity and spin-polarization, respectively. (**c**) Photoemission intensity obtained by integrating within the boxes for the USR and TSS, showing a substantial increase in population from the pump excitation. (**d**) The delay-dependent spin polarization of both regions, which is constant within statistical error bars.

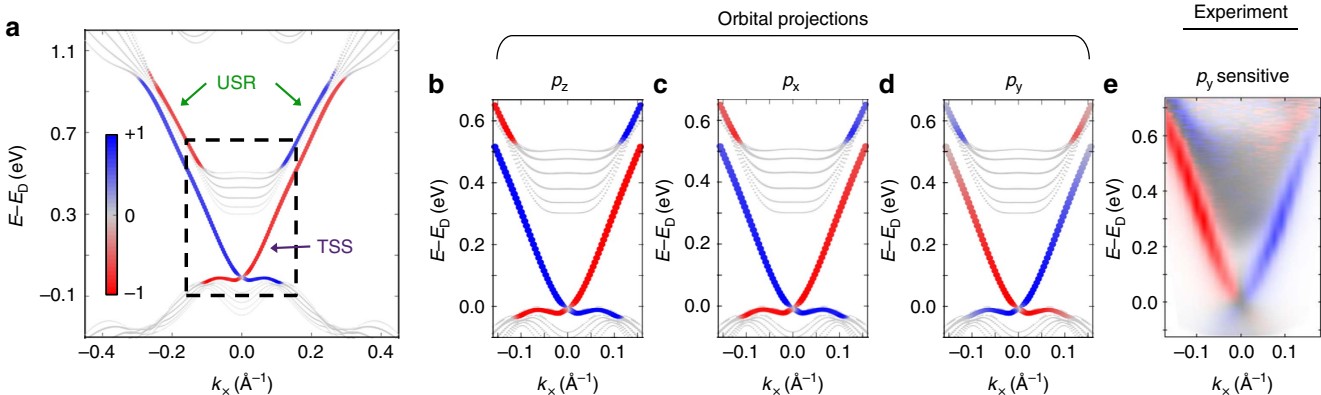

**Figure 4 | Density functional theory calculations of the spin resolved electronic structure along Γ-K. (a)** The marker sizes denote the magnitude of the projection of each state onto the top quintuple layer of the slab, whereas the colors denote the spin-polarization as indicated by the colorscale. Small gray markers therefore show unpolarized bulk-like bands, while large blue/red markers show surface-localized bands, such as the topological surface state (TSS) and unoccupied surface resonance (USR), with spin polarization along the $k_y$ axis. The energy scale is referenced to the Dirac point energy, $E_D$. **(b–d)** Same as previous panel, projected onto $p_z$, $p_x$, and $p_y$ orbitals, respectively. **(e)** The experiment was performed with $s$-polarized probe photons, and therefore can be compared to the $p_y$ projected calculation.

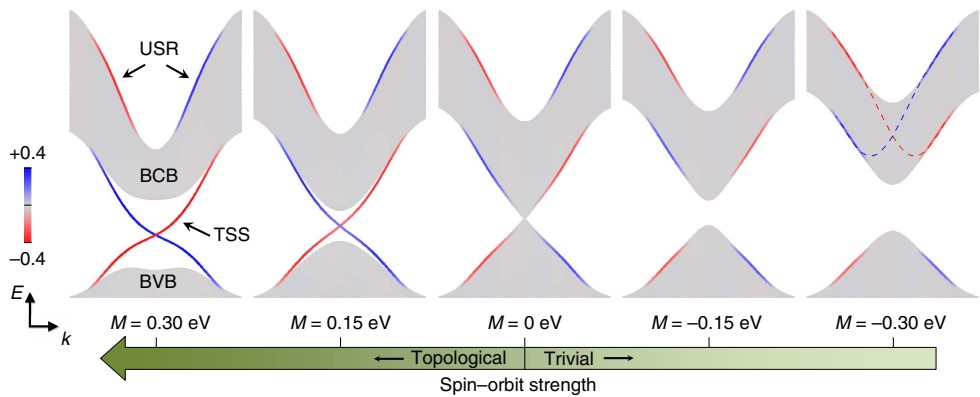

**Figure 5 | Tight binding model calculation results.** The colors represent the magnitude of the $y$-component of the polarization, so the unpolarized bulk-like bands appear in gray. The calculation on the left ($M = 0.3$ eV) qualitatively captures the experimental observations of the topological surface state (TSS) and unoccupied surface resonance (USR). Moving right, the mass term $M$ in the calculation is decreased to model a reduction in the spin-orbit interaction. These results show that the observed TSS and USR evolve into a trivial Rashba-pair through the process of band inversion. The dashed lines for $M = -0.3$ eV are a guide to the eye overlaid to show that the Kramer's point of the Rashba-pair is buried in the bulk conduction band (BCB).

quasiparticle inference patterns observed by scanning tunneling microscopy[22]. The USR was also identified in a STARPES measurement of $Bi_2Se_3$, where it led to a reversal of the measured spin-polarization in the unoccupied states[23]. Our observations provide the first direct measurement of the full momentum dependent spin texture of the USR and its entangled spin-orbit texture. As we now discuss, these observations may have important implications for understanding the structure of the TSS itself.

**Tight binding model of the topological phase transition.** Insight into the relationship between the USR and TSS, such as the anticorrelation of the spin-textures of the two, can be pursued by examining the evolution of the surface states/resonance as the system goes through a trivial to non-trivial topological phase transition (triggered by band inversion). To do so, we adopt the tight-binding model of refs 2,34. We find that the addition of a phenomenological term to the Hamiltonian of the standard surface Rashba form (see Methods) qualitatively reproduces the USR: a spin-polarized surface-localized state split above the BCB,

with the USR's characteristic reduction in strength towards Γ and its spin-polarization oriented opposite to that of the TSS as shown on the far left of Fig. 5. Furthermore, this model shows the continuation of the TSS extending far above the gap, and well outside the momentum region of band-inversion, where it is accompanied by the USR.

We can then gradually change the mass term $M$ from positive (topologically non-trivial) to negative (trivial) and watch the evolution of the surface states, shown from left to right in Fig. 5. The system undergoes a topological phase transition at $M = 0$ where the bulk band gap closes. On the trivial side of the transition one sees that the bands previously associated with the TSS no longer cross the gap, and along with the USR bands, become similar to a canonical pair of Rashba-split bands. The Kramer's point of this pair is buried in the middle of the BCB, as indicated by the guide to the eye in the far-right panel, and so is not observed. In this scenario, we can understand why the spin textures of the TSS and USR are opposite—because they originate from a pair of Rashba-split bands. In fact, recent experiments provide strong evidence for the existence of such states on the trivial side of topological phase transitions[35,36].

## Discussion

Overall, we find qualitative agreement between our tight-binding results and the DFT calculations and experimental data, which suggests there exist Rashba-split bands on the trivial side of the transition, which upon inversion, evolve into the TSS and USR. In Supplementary Fig. 6 we show that other phenomenological perturbations to the basic tight-binding model[2,34], such as an electrostatic surface potential, produce qualitatively similar behavior. These provide compelling cases for an intrinsic relationship between the TSS and USR and their respective spin-textures and may suggest that the existence of the USR is a robust phenomenon in topological insulators such as $Bi_2Se_3$.

This work exemplifies the widespread influence of the SOI in determining the spin and electronic structure of materials. From a practical perspective, the characterization of the rich spin-dependent electronic structure of a prototypical topological insulator is critical for furthering realistic applications of these exciting materials. On a more fundamental level, our direct visualization of the complete spin-polarized electronic structure reveals that the SOI has previously unrecognized consequences for the states affected by band inversion, even beyond the formation of the TSS. Considerable advances in understanding these systems could be made through more sophisticated calculations and systematic experiments performed as a function of SOI strength[35]. The continued use and development of STARPES will be key in furthering this and related fields due to its unique capability of directly mapping spin structures, both below and above $E_F$, in and out of equilibrium.

## Methods

**Photoemission experiments.** The STARPES experiments were performed using a high-efficiency and high-resolution spin-resolved photoelectron spectrometer[20,21]. Ultrafast laser pulses are provided by a cavity-dumped Ti:Sapphire oscillator operating at a repetition rate of $\sim 3.6$ MHz. The pump and probe pulses consist of the fundamental output at a photon energy of 1.5 eV, and the fourth harmonic at 6.0 eV, respectively. The length of the pump path is controlled by a mechanical translation stage to vary the delay between the arrival of the pump and probe pulses on the sample surface. All measurements are performed with an incidence pump fluence of $\sim 50\,\mu J\,cm^{-2}$. The energy and momentum resolutions are 15 meV and $\sim 0.02\,\text{Å}^{-1}$. The $Bi_2Se_3$ single crystals were grown by directional slow solidification in an inclined ampoule and cleaved in-situ along the (111) plane in vacuum of $\sim 5 \times 10^{-11}$ torr. All measurements were taken at a sample temperature of $\sim 80$ K.

**Density functional theory calculations.** DFT calculations were performed using the Vienna Ab-initio Simulation Package (VASP)[37–40] using projector augmented wave[41,42] Perdew-Burke-Ernzerhof[43,44] GGA-type pseudopotentials. We used a slab geometry with 7 quintuple layers with a 30 Å vacuum layer using experimental coordinates for the lattice. The results were cross-checked with calculations for 6 and 8 quintuple layers. The calculations were performed on a $15 \times 15 \times 1$ $\Gamma$-centered momentum grid with a wave function cutoff of 250 eV. Spin-orbit coupling is included in a non-self-consistent calculation after the density is determined.

**Tight binding calculations.** The tight-binding calculations are performed based on a minimal model for a 3D topological insulator with four orbitals per site[2,34]:

$$\hat{H}_{TI} = \sum_{n,\mathbf{k}_\parallel} \left\{ \mathbf{c}_{n,\mathbf{k}_\parallel}^\dagger \left( \frac{B}{a^2}\boldsymbol{\Gamma}_0 - i\frac{A}{2a}\boldsymbol{\Gamma}_3 \right) \mathbf{c}_{n+1,\mathbf{k}_\parallel} + \text{H.c.} + \right.$$
$$\left. \mathbf{c}_{n,\mathbf{k}_\parallel}^\dagger \left[ C\mathbf{1} + d(\mathbf{k}_\parallel)\boldsymbol{\Gamma}_0 + \frac{A}{a}\left( \boldsymbol{\Gamma}_1 \sin k_x a + \boldsymbol{\Gamma}_2 \sin k_y a \right) \right] \mathbf{c}_{n,\mathbf{k}_\parallel} \right\} \tag{1}$$

The summation runs over momentum $\mathbf{k}_\parallel$ and layer index $n$. The operator $\mathbf{c}_{n,\mathbf{k}_\parallel}$ annihilates an electron in the basis $\left( |P1_z^+, \uparrow\rangle, |P1_z^+, \downarrow\rangle, |P2_z^-, \uparrow\rangle, |P2_z^-, \downarrow\rangle \right)$ as described in ref. 2. $d(\mathbf{k}_\parallel) = M - 2B/a^2 + 2B/a^2(\cos k_x a + \cos k_y a - 2)$, $\boldsymbol{\Gamma}_i$ are $4 \times 4$ Dirac matrices, and $\mathbf{1}$ is the $4 \times 4$ identity operator. The numerical values of the parameters used are $A = 0.375a$ eV, $B = 0.25a^2$ eV, and $C = 3.0$ eV, which determines the overall energy offset. The parameter $M$ determines the sign and magnitude of the band-gap. For the calculations in this manuscript, $M$ is tuned from $-0.3$ eV (trivial bulk band structure) to $+0.3$ eV (inverted bulk bands).

We phenomenologically investigate the effect of surface perturbations of different forms: $\hat{H}_{total} = \hat{H}_{TI} + \hat{H}_{Rashba} + \hat{H}_{SP}$, where:

$$\hat{H}_{Rashba} = R \sum_{n=\{0,N\},\mathbf{k}_\parallel} \mathbf{c}_{n,\mathbf{k}_\parallel}^\dagger \left( \sigma_y \sin k_x a - \sigma_x \sin k_y a \right) \mathbf{c}_{n,\mathbf{k}_\parallel} \tag{2}$$

$$\hat{H}_{SP} = SP \sum_{n=\{0,N\},\mathbf{k}_\parallel} \mathbf{c}_{n,\mathbf{k}_\parallel}^\dagger \mathbf{1} \mathbf{c}_{n,\mathbf{k}_\parallel} \tag{3}$$

Here $\hat{H}_{Rashba}$ models a surface Rashba effect with magnitude $R$, and $\hat{H}_{SP}$ models an electrostatic surface potential with magnitude $SP$. $\sigma_i$ are the Pauli matrices. Note that the summation includes only the outermost top ($n = 0$) and bottom ($n = N$) layers of the slab. The calculations are performed with $N = 100$ layers. While the calculations in the main text do not include $\hat{H}_{SP}$, its effect is considered in Supplementary Fig. 6.

**Data availability.** The data that support the findings of this study are available from the corresponding author upon request.

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

## Acknowledgements

We thank A. Bostwick for significant help with software development, and P.S. Kirchmann and S.-L. Yang for meaningful discussions. This work was mainly supported by the Director, Office of Science, Office of Basic Energy Sciences, Materials Sciences and Engineering Division, of the U.S. Department of Energy, under Contract No. DE-AC02-05CH11231, as part of the Ultrafast Materials Science Program (KC2203). C.J. and Z.H. were supported by the Advanced Light Source, which is supported by the Director, Office of Science, Office of Basic Energy Sciences, of the U.S. Department of Energy under Contract No. DE-AC02-05CH11231. J.A.S. and Z.-X.S. were primarily supported by the Director, Office of Science, Office of Basic Energy Sciences, Materials Sciences and Engineering Division, of the U.S. Department of Energy, under Contract No. DE-AC02-76SF00515, and in part by the Gordon and Betty Moore Foundation's EPiQS Initiative through Grant GBMF4546. Sample growth and theoretical work were supported by the Director, Office of Science, Office of Basic Energy Sciences, Materials Sciences and Engineering Division, of the U.S. Department of Energy, under Contract No. DE-AC02-05CH11231 as part of the Quantum Materials Program (KC2202). A.F.K. was in part supported by the Laboratory Directed Research and Development Program of Lawrence Berkeley National Laboratory under U.S. Department of Energy Contract No. DE-AC02-05CH11231.

## Author contributions

C.J. and J.A.S developed the experimental optics system and devised the experiments. C.J., J.A.S. and K.G. carried out the experiment. C.J. and J.A.S analyzed the experimental data. Calculations were performed by A.F.K., D.-H.L., and J.A.S. Samples were prepared by C.R.R. and R.J.B. Z.H., Z.-X.S., and A.L. were responsible for experiment planning and infrastructure. All authors contributed to the interpretation and writing of the manuscript.

## Additional information

**Competing financial interests:** The authors declare no competing financial interests.

