## [Peer Review File · Nature Communications]

REVIEWERS' COMMENTS:

Reviewer #1 (Remarks to the Author):

The article "Spin-Polarized Surface Resonances Accompanying Topological Surface State Formation" reports about a time, spin and angle resolved investigation of photoexcited topological insulator Bi₂Se₃. The data are of very high quality and reveal new aspects on the spin polarized band structure of topological insulators. By analyzing transiently occupied states that are located around 0.4 eV above the Fermi level, the authors could identify surface resonances with strong spin polarization. The authors measured the spin component lying in plane and orthogonal to the electron wavevector. By these means, they show that states of the Dirac cone and surface resonance hold opposite spin polarizations. Such results have been modeled by ab-initio calculation and can be easily interpreted. The surface resonances are the remnant portions of Rashba branches. Within this picture the Dirac cone emerges as a result of an hypothetical tuning of the spin-orbit coupling. Dirac states can be viewed as non-trivial entities arising from original Rashba branches and connecting the valence and conduction band once spin-orbit coupling is strong enough to induce parity inversion at the time reversal invariant moments.

This article provides a novel perspective on the electronic structure and spin texture of topological states. It is a nice example on how spin and angle spectroscopy of the excited state can provide a wavevector and spin mapping in electronic states that are not accessible in equilibrium conditions. The presentation is clear and explains in detail the experiment and the modeling. I recommend publication of the present version of the manuscript in Nature Communications.

Reviewer #2 (Remarks to the Author):

In the manuscript NCOMMS-16-15709-T "Spin-Polarized Surface Resonances Accompanying Topological Surface State Formation" C.Jozwiak et al present spin-, time-, and angle-resolved photoemission spectroscopy measurement results for topological insulator Bi₂Se₃ accompanied by DFT and tight-binding calculations.

Having these high efficiency measurements they find unoccupied spin-polarized surface resonance and give very convincing interpretation using Rashba surface states.

The manuscript is clearly written and results are well argued. Previous research activity in this field is correctly cited. The paper can be interesting for physics and materials science communities and I recommend it for publication in the Nature Communications without further modification.